# Supplementing Ryegrass Ameliorates Commercial Diet-Induced Gut Microbial Dysbiosis-Associated Spleen Dysfunctions by Gut–Microbiota–Spleen Axis

**DOI:** 10.3390/nu16050747

**Published:** 2024-03-05

**Authors:** Qasim Ali, Sen Ma, Boshuai Liu, Jiakuan Niu, Mengqi Liu, Ahsan Mustafa, Defeng Li, Zhichang Wang, Hao Sun, Yalei Cui, Yinghua Shi

**Affiliations:** 1Department of Animal Nutrition and Feed Science, College of Animal Science and Technology, Henan Agricultural University, Zhengzhou 450046, China; qasimali@henau.edu.cn (Q.A.); ms0321021@163.com (S.M.); boshuailiu1@126.com (B.L.); niujiakuan2022@126.com (J.N.); 2019110376@sdau.edu.cn (M.L.); leadephone@126.com (D.L.); zcwang@henau.edu.cn (Z.W.); sunhao@henau.edu.cn (H.S.); yaleicui@henau.edu.cn (Y.C.); 2Henan Key Laboratory of Innovation and Utilization of Grassland Resources, Zhengzhou 450002, China; 3Henan Herbage Engineering Technology Research Center, Zhengzhou 450001, China; 4Department of Animal Nutrition, Sichuan Agricultural University, Chengdu 611130, China; dr.ahsan.mustafa@gmail.com

**Keywords:** ryegrass, geese, gut microbiota, systemic inflammation, short-chain fatty acids, gut–microbiota–spleen axis

## Abstract

The type and composition of food strongly affect the variation and enrichment of the gut microbiota. The gut–microbiota–spleen axis has been developed, incorporating the spleen’s function and maturation. However, how short-chain fatty-acid-producing gut microbiota can be considered to recover spleen function, particularly in spleens damaged by changed gut microbiota, is unknown in geese. Therefore, the gut microbial composition of the caecal chyme of geese was assessed by 16S rRNA microbial genes, and a Tax4Fun analysis identified the enrichment of KEGG orthologues involved in lipopolysaccharide production. The concentrations of LPS, reactive oxygen species, antioxidant/oxidant enzymes, and immunoglobulins were measured from serum samples and spleen tissues using ELISA kits. Quantitative reverse transcription PCR was employed to detect the Kelch-like-ECH-associated protein 1–Nuclear factor erythroid 2-related factor 2 (Keap1-Nrf2), B cell and T cell targeting markers, and anti-inflammatory/inflammatory cytokines from the spleen tissues of geese. The SCFAs were determined from the caecal chyme of geese by using gas chromatography. In this study, ryegrass-enriched gut microbiota such as *Eggerthellaceae*, *Oscillospiraceae*, *Rikenellaceae*, and *Lachnospiraceae* attenuated commercial diet-induced gut microbial alterations and spleen dysfunctions in geese. Ryegrass significantly improved the SCFAs (acetic, butyric, propionic, isovaleric, and valeric acids), AMPK pathway-activated Nrf2 redox signaling cascades, B cells (*B220*, *CD19*, and *IgD*), and T cells (*CD3*, *CD4*, *CD8*, and *IL-2*, with an exception of *IL-17* and *TGF-β*) to activate anti-inflammatory cytokines (*IL-4* and *IL-10*) and immunoglobulins (IgA, IgG, and IgM) in geese. In conclusion, ryegrass-improved reprogramming of the gut microbiota restored the spleen functions by attenuating LPS-induced oxidative stress and systemic inflammation through the gut–microbiota–spleen axis in geese.

## 1. Introduction

Food composition and its intake duration largely impact the enrichment and diversity of the intestinal microbiota [1]. On the other hand, a higher utilization of saturated fats, salts, animal protein sources, and sugars damages beneficial bacteria, leading to substantial alterations in gut microbiota richness and gut barrier dysfunctions. However, the use of plant protein sources and dietary fibers may be linked with an increased number of beneficial bacteria [2]. Dietary fat, to varying degrees, can cause oxidative stress in the gut’s epithelial cells [3]. In addition, a high-fat diet enriches lipopolysaccharide (LPS)-producing bacteria, particularly *Proteobacteria* and *Firmicutes* [4]. Surprisingly, these gut microbial alterations can be restored by high-dietary-fiber-source diets [5].

The production of oxidative stress due to altered gut microbial LPS activation may lead to systemic inflammation [2,6,7], metabolic disorders, and inflammatory diseases, including type II diabetes, sepsis, splenectomy, etc. [8,9]. The spleen is an immune organ that elicits hematological functions [10]. Animals with damaged spleens from oxidative stress [11], chronic stress [12], and LPS-induced stress [13] show compromised immune functions because the spleen is responsible for removing aged and damaged erythrocytes and bacteria, recycling iron, and producing antibodies [14,15]. The gut microbiota, upon maturing the spleen [16,17], controls its functions, such as those involved in systemic inflammatory processes [17,18,19]. 

Different possible modifiable factors such as alginate oligosaccharides [19], fine particulate matter [20], feed antibiotics [5], dietary fiber [21,22], immunoglobulin G [23], berry pomaces and phenolic-enriched extractives [24], astragalus polysaccharides [25], oat beta-glucan [26], and *Lactobacillus acidophilus*, *Lactobacillus reuteri*, and *Lactobacillus salivarius* probiotic strains [27] have been used in animal models to deal with spleen dysfunctions. In addition, Guelph-resistant (Gu-R) and single-comb (S.C.) strains have been injected in domestic chickens to treat splenomegaly [28]. Moreover, ryegrass shows an anti-inflammatory effect; however, the exact mechanisms by which ryegrass attenuates gut microbial LPS-induced systemic inflammation in spleen organs are still unknown in geese. 

Geese are naturally grazing animals, and due to their special capacity to consume high-fiber feeds, pasture is recommended to improve their health and save grain feed [29]. The current poultry industry is established on cereals with lower dietary fiber contents [30], which may lead to higher growth performance and compromised digestive health. However, these improvements are dangerous for visceral growth and may cause low immunity, intestinal infections, and metabolic diseases [31]. Recently, we have investigated the useful impacts of ryegrass on regulating the gut microbiota, antioxidant, antimicrobial, and anti-inflammatory characteristics in Wanfu geese [2]. Based on previous reports that the gut–microbiota–spleen axis modulates systemic inflammation-related processes [18,19,32], we aimed to investigate the effects of ryegrass on attenuating the gut microbial alteration-associated spleen dysfunctions in geese. In this context, first, we developed a mechanistic pathway in which ryegrass, upon restoring SCFA-producing gut microbiota, activates the Nrf2 redox signaling pathway in the spleen organs of geese. Then, it was investigated that this pathway elicits B cells and regulatory T cells to stimulate anti-inflammatory cytokines and immunoglobulins, which alternatively attenuate oxidative stress-induced systemic inflammation in geese. Lastly, a correlation analysis of the gut microbiome with host markers was established to better understand the spleen functions through the gut–microbiota–spleen axis.

## 2. Materials and Methods

### 2.1. Ethical Approval Statement

The animal experiments were approved by the Research Bioethics Committee of the Henan Agricultural University.

### 2.2. Animals, Diets, and Management

Wanfu mixed-sex geese (*n* = 180, 1 day old) were sourced from the Henan Daidai Goose Agriculture and Animal Husbandry Development Co., Ltd. (Zhumadian, China). Reaching the age of 25 days, geese with similar body weights were assigned into two groups, i.e., (1) the in-house feeding group (IHF, *n* = 90) and (2) the artificial pasture grazing group (AGF, *n* = 90). Geese in each group were divided into six replicates of 15 birds, each with an ad libitum supply of feed and water. The in-house feeding group was provided with commercial diets (two stages). The artificial pasture grazing group was fed ryegrass (6:00 a.m. to 6:00 p.m.) with a commercial diet (only once a day, 7:00 p.m.). The nutrients and chemical composition of the diet offered during the trial are shown in Appendix A. The trial duration was 66 days (Appendix A).

### 2.3. Sample Collection

At the ages of 45 d, 60 d, and 90 d, six geese were slaughtered from each group by the halal method. The average daily feed intake and live body weight gain were measured weekly. From the jugular vein of the geese, blood samples were collected in sterile blood vessels, which were then centrifuged at 4000× *g* for 15 min at 4 °C, and the serum samples were preserved at −80 °C for further examinations. The fresh caecal chyme was taken into sterile centrifuge tubes and stored at −80 °C. The spleen and caecal tissues were washed with phosphate-buffered saline, kept in liquid nitrogen, and then stored at −80 °C for further examination.

### 2.4. Determination of LPS, ROS, IgA, IgG, and IgM

The concentrations of LPS and ROS were measured from the serum samples and spleen tissues, while the values of IgA, IgG, and IgM were determined from the serum samples and spleen and caecal tissues. All of the kits were bought from Shanghai Enzyme Link Biotechnology Co., Ltd. (Shanghai, China), and the experimental protocols were carried out following the manufacturer’s directions.

### 2.5. Determination of Antioxidant Activity

Using ELISA kits (Shanghai Meilian Biology Technology, Shanghai, China), the levels of glutathione reductase (GSR) and heme oxygenase 1 (HO-1) in splenic tissues were assessed. Total superoxide dismutase (T-SOD, #A001-1), malondialdehyde (MDA, #A003-1), glutathione peroxidase (GSH-Px, #A005), catalase (CAT, #A007-1), and total antioxidant capacity (T-AOC, #A015-2–1) were determined from spleen tissues by applying diagnostic kits (Nanjing Jiancheng Bioengineering Institute, Nanjing, China) based on the manufacturer’s directions.

### 2.6. RNA Extraction and Gene Expression Analysis

Total RNA from caecal and spleen tissues was collected and reverse-transcribed into cDNA using HiScript^®^ III RT SuperMix for qPCR (+gDNA wiper) (Vazyme, Nanjing, China). The specific primers of genes produced by Primer3Web version 4.1.0 (https://primer3.ut.ee/ (accessed on 3 March 2023)) are listed in Appendix A. The RT-qPCR was carried out using ChamQ Universal SYBR qPCR Master Mix from Vazyme Biotechnology (Nanjing, China) on a C1000 Touch PCR Thermal Cycler (BIO-RAD Laboratories, Shanghai, China). It was executed under the following conditions: 40 cycles of 95 °C for 15 s and 60 °C for 30 s. The messenger ribonucleic acid (mRNA) expression of target genes relative to beta-actin was estimated using the 2^−ΔΔCT^ approach [33].

### 2.7. Determination of Short-Chain Fatty Acids

The in vivo testing for the concentration of short-chain fatty acids in caecal chyme was assessed using gas chromatography (GC), as revealed by Liu et al. [34]. 

### 2.8. DNA Extraction and 16S rRNA Gene Sequencing

The microbial community genomic DNA and the bacterial 16S rRNA gene sequencing were determined using the method described by Ali et al. [2].

### 2.9. Illumina MiSeq Sequencing

The Illumina MiSeq technology (Illumina, San Diego, CA, USA) was utilized to pool purified amplicons and perform paired-end sequencing (2 × 300) under standard protocols provided by Majorbio Bio-Pharm Technology Co., Ltd. (Shanghai, China). 

### 2.10. Bioinformatics Analysis of Sequencing Data

The raw 16S rRNA gene sequencing reads were demultiplexed, quality-filtered by Trimmomatic, and merged by FLASH according to the method described by Ali et al. [2]. Alpha diversity was calculated based on the OTU profiles from the MOTHUR program (version v.1.30.2, https://mothur.org/wiki/calculators/ (accessed on 10 July 2023)). For alpha diversity, the Kruskal–Wallis H test was applied to determine the Ace index, Chao index, Shannon index, Simpson index, and Sobs index among the six groups. The false discovery rate (FDR) was considered to adjust *p* values. NMDS-based β-diversity analysis by the Bray–Curtis dissimilarity matrix was employed to determine species composition between the samples. A stamp variance analysis was applied to analyze the taxonomic profiles of gut microbes (http://www.cloudtutu.com/#/stamp (accessed on 17 July 2023)). Tax4Fun v.0.3.1 (http://tax4fun.gobics.de/ (accessed on 20 July 2023)) was utilized for inferred metagenome profiling against canonical pathways of the Kyoto Encyclopedia of Genes and Genomes (KEGG). 

### 2.11. Statistical Analysis

The raw data are expressed as the mean ± SD. Statistical analyses were accomplished using the SPSS 20.0 software (=D3 SPSS, Inc., 2009, Chicago, IL, USA, www.spss.com (accessed on 12 April 2023)). The difference between the two groups was found by Student’s *t*-test, and *p* < 0.05 was considered statistically significant. GraphPad Prism (version 8.3.0.) was used to make the graphs, and the Spearman correlation analysis was achieved using Majorbio (https://cloud.majorbio.com (accessed on 13 August 2023)) and OECloud Tools (https://cloud.oebiotech.cn (accessed on 17 August 2023)) to evaluate the host markers’ relationships. 

## 3. Results

### 3.1. Commercial Diet Caused Gut Microbial Alterations in Geese

To study the impacts of ryegrass on the geese’s gut microbiota, we performed high-throughput amplicon sequencing of 16S rRNA microbial genes from geese in the IHF and AGF groups. To determine the diversity, richness, and evenness of the gut microbiota of geese in the IHF and AGF groups, α-diversity was determined by applying the Ace index, Chao index, Shannon index, Simpson index, and Sobs index. We noticed a considerable decline in the α-diversity of the IHF group compared to that of the AGF group (Appendix A). The NMDS-based β-diversity analysis based on the Bray–Curtis dissimilarity matrix revealed significant modifications in caecal microbiota at 45d (Stress: 0.069, *R* = 0.754, *p* = 0.003), 60d (Stress: 0.087, *R* = 0.387, *p* = 0.003), and 90d (Stress: 0.049, *R* = 0.274, *p* = 0.059) between the IHF and AGF geese groups (Appendix A). The microbial and bacterial taxonomies of the caecal microbiota were equated between the two groups at the class and family levels (Figure 1A). Equating the percentages of bacteria at the class level, ryegrass significantly reduced the *Clostridia*, *Bacteroidia*, *Gammaproteobacteria*, *Desulfovibrionia*, and *Negativicutes* compared to those of the IHF group (Figure 1A), while at the family level, the percentages of *Ruminococcaceae*, *norank_o__RF39*, *Lachnospiraceae*, *norank_o__Clostridia_UCG-014*, *Eubacterium_coprostanoligenes_group*, and *Erysipelatoclostridiaceae* were more reduced in the AGF group than in the IHF group (Figure 1A). 

Next, we examined detailed alterations in the caecal microbiota at various taxonomic levels in the AGF group compared to that of the IHF group. At the class level, the AGF group had the greatest abundances of *Bacilli*, *Coriobacteriia*, *Actinobacteria*, *Vampirivibrionia*, *Verrucomicrobiae*, and *Alphaproteobacteria* relative to the IHF group (Figure 1B and Appendix A). In contrast, *Clostridia*, *Bacteroidia*, *Gammaproteobacteria*, *Desulfovibrionia*, and *Negativicutes* were absent in the AGF group compared to those in the IHF group (Figure 1B and Appendix A). The abundance of 40 bacterial species at the family level was detected to differ between the two groups, of which 20, 18, and 20 were significantly lowered after ryegrass supplementation at 45 d, 60 d, and 90 d, respectively (Figure 1C and Appendix A). Interestingly, at 45 d, five species—*norank_o__Clostridia_vadinBB60_group*, *Akkermansiaceae*, *norank_o__Bacteroidales*, *Coriobacteriales_Incertae_Sedis*, and *Enterococcaceae*—and at 60 d and 90 d, three species—*Planococcaceae*, *Bifidobacteriaceae*, and *Lactobacillaceae*—were abundantly higher in the AGF group than in the IHF group (Figure 1C and Appendix A). 

### 3.2. Commercial diet-Dependent Gut Microbial Alterations Are Paralleled by LPS-Induced Oxidative Stress Production

The metagenome-predicted functional modifications of caecal microbial KEGG terms were detected by applying Tax4Fun to 36 samples of geese. A total of 6, 40, and 257 KEGG terms were identified at KEGG levels 1, 2, and 3, respectively. To recognize which strains aid in LPS biosynthesis, we concentrated on KEGG terms at level 3 (Figure 2A) and gut microbiota at the family level (Figure 2B). By applying a Spearman correlation analysis between the LPS-synthesizing strains and gut microbiota, we detected that *Eubacterium_coprostanoligenes_group*, *Norank_o__Clostridia_UCG-014*, and *Erysipelatoclostridiaceae* were the main bacteria that were positively correlated with pertussis, the bacterial invasion of epithelial cells, Vibrio cholera infection, and Salmonella infection (Figure 2B and Appendix A). Most of the highly abundant bacteria in the IHF group were involved in the bacterial invasion of epithelial cells. Interestingly, the highly prevalent bacteria in the ryegrass intake group (i.e., *Oscillospiraceae* and *Bacteroidaceae*) were involved in reducing pertussis, the bacterial invasion of epithelial cells, Vibrio cholera infection, and Salmonella infection (Figure 2B and Appendix A). 

Comparing the LPS-causing bacterial strains with the concentration of LPS in the caecum and spleen tissues, we hypothesized that diet-dependent LPS production could disturb spleen functions by generating oxidative stress in geese. We first confirmed this by establishing the LPS/NADPH-induced ROS mechanism on the reliance on dietary intervention. In this mechanism, the concentration of LPS measured using the ELISA kits was increased in the spleen tissues and serum samples of the IHF group (Figure 2C and Appendix A). LPS plays a pivotal role in activating the regulatory subunit gene (*p47^phox^*) of nicotinamide adenine dinucleotide phosphate (NADPH) by augmenting proto-oncogene tyrosine-protein kinase (*Srck*) and phosphatidylinositol-3-kinase (*PI3K*). Indeed, we confirmed this by measuring the mRNA expression levels of *Srck*, *PI3K*, and *p47^phox^* that were surprisingly more increased in the spleen tissues of the IHF group than in the AGF group at 45 d, 60 d, and 90 d (Figure 2D–F). Next, we sought to identify whether the increased *NADPH* allows LPS to connect with Toll-like receptors (TLRs) and then promotes the myeloid differentiation primary response 88 (MyD88) pathway. Herein, first, we determined the mRNA expression levels of *NADPH*, *TLR2*, and *TLR4* from the spleen tissues of geese. We found that the expression levels of *NADPH*, *TLR2*, and *TLR4* were increased in ryegrass-lacking geese (Figure 2G–I). Then, we detected a higher mRNA level of *MyD88* in the IHF group than in the AGF group (Figure 2J), which suggested that a gradual increase in *NADPH* may permit LPS to bind with *MyD88*. The NADPH/MyD88 activation may promote ROS production. Intriguingly, the splenic and serum ROS production was paralleled by increased *NADPH* and *MyD88* in the IHF group compared to those of the AGF group (Figure 2K and Appendix A).

### 3.3. Commercial Diet-Induced Oxidative Stress Modulates Spleen Weight and Splenocyte Populations via NF-κB/NLRP3 Pathway-Induced Systemic Inflammation

Given that the oxidative stress-induced NF-κB pathway exacerbates NLR family pyrin domain containing 3 (NLRP3) activity, which influences systemic inflammation through regulating inflammatory cytokines, we hypothesized that its regulation could affect splenocyte populations. First, we developed a priming signal pathway for NLRP3 activation and hypothesized that the regulation of NF-κB could be MyD88-dependent TRAF6 activation in geese. 

We detected higher mRNA expression levels of *MyD88* (Figure 2J), interleukin-1 receptor (IL-1R)-associated kinase (*IRAK*), TNF receptor-associated factor (*TRAF6*), and transforming growth factor β-activated kinase 1 (*TAK1*) in the IHF group compared with the AGF group (Figure 3A–C). The increased mRNA expression level of *TAK1* suggested that it must activate IκB kinase α (*IKKα*) (Figure 3D) to degrade NF-κB inhibitor *IκBα* (Figure 3E) and, in turn, activate *NF-κB*-facilitated *NLRP3* in IHF geese as compared to AGF geese (Figure 3F,G). In response to *NLRP3* activation, *caspase-1* was increased in the IHF group which further influenced the pro-inflammatory cytokines interleukin 1 beta (*IL-1β*), *IL-18*, inducible nitric oxide synthase (*iNOS*), cytochrome c oxidase subunit 2 (*COX2*), tumor necrosis factor-alpha (*TNF-α*), and *IL-6* in the spleen tissues of geese (Figure 3H–N).

How NLRP3-induced systemic inflammation affects splenocyte populations needs to be explored in geese. To obtain insight into splenocyte population detection, we first showed the effect of inflammation on spleen weight, length, and width. Contrary to the IHF group, ryegrass resulted in a significant reduction in spleen weight, the ratio of spleen to body weight, and spleen length and width (Figure 4A and Appendix A). To investigate the splenocyte populations related to the AGF-induced decrease in spleen weight, length, and width, we then identified the mRNA expression levels of B cells (B220, CD19, and IgD) and T cells (CD3, CD4, CD8, IL-2, IL-17, and TGF-β). Of note, the mRNA expression levels of *B220*, cluster of differentiation 19 (*CD19*), immunoglobulin D (*IgD*), *CD3*, *CD4*, *CD8*, and interleukin 2 (*IL-2*) in the AGF group were higher than in the IHF group (Figure 4B–H). In contrast, the mRNA levels of *IL-17* and transforming growth factor-β (*TGF-β*) were significantly decreased in the AGF group compared with the IHF group (Figure 4I,J). Positive correlations were detected between the expression levels of *IL-17*, *TGF-β*, spleen weight, spleen relative weight, and spleen length and width. On the contrary, a negative correlation was found between the expression levels of *B220*, *CD19*, *IgD*, *CD3*, *CD4*, *CD8*, *IL-2*, spleen weight, spleen relative weight, and spleen length and width (Figure 4K). 

### 3.4. Long-Term Ryegrass Intake Causes Gut Microbial Short-Chain Fatty Acid Enrichment

The production of SCFAs by the gut microbiome plays an important role in controlling myeloid and lymphocyte cell populations to induce immune molecules that support balanced immunity [35,36]. We determined the concentrations of SCFAs through gas chromatography from the caecal chyme of geese (Figure 5A–E). The levels of butyric, propionic, and acetic acids were dominant over valeric and isovaleric acids in the AGF group compared to the IHF group. To determine the potential associations between the relative bacterial abundance and SCFA levels in caecal chyme samples, we developed a Spearman correlation analysis. There were significant positive correlations of the enrichment of *Alphaproteobacteria* with acetic acid (*R* = 0.895, *p* < 0.001), propionic acid (*R* = 0.866, *p* < 0.001), valeric acid (*R* = 0.888, *p* < 0.001), isovaleric acid (*R* = 0.743, *p* < 0.001), and butyric acid (*R* = 0.661, *p* < 0.001) at the class level (Figure 5F and Appendix A). In contrast, a significant positive correlation of *Bacteroidaceae* with acetic acid (*R* = 0.912, *p* < 0.001), propionic acid (*R* = 0.920, *p* < 0.001), butyric acid (*R* = 0.793, *p* < 0.001), isovaleric acid (*R* = 0.764, *p* < 0.001), and valeric acid (*R* = 0.865, *p* < 0.001) was observed at the family level (Figure 5F and Appendix A). 

### 3.5. Ryegrass-Dependent SCFA Regulation Improved Keap1-Nrf2 Pathway Activation

Gut microbial SCFAs play a pivotal role in the activation of the Keap1-Nrf2 redox signaling pathway by patrolling the AMPK pathway [37]. How SCFAs activate the redox signaling pathway is not described in geese. We hypothesized that the higher expression of SCFA receptors could activate the AMPK pathway and then collectively inhibit HDAC. Upon HDAC inhibition, Keap1 dissociates from Nrf2 and then enters the activation of antioxidant-responsive element (ARE)-dependent genes. With this agreement, we first identified that the mRNA expression levels of *GPR109A*, *FFAR2*, *FFAR3*, and *AMPK* were increased (Figure 6A,B) and the expression levels of *HDAC* declined in the spleen tissues of AGF geese (Figure 6C). This suggests that the HDAC inhibition reduced the Nrf2 suppressor *Keap1* mRNA expression (Figure 6D) and then separated Keap1/Nrf2 in augmenting the Nrf2 encoding gene *NFE2L2* (Figure 6E) and *Nrf2* itself (Figure 6F).

The activation of the Nrf2 pathway is considered to induce cellular redox homeostasis, and any defect in Nrf2 activity is known to change the cellular defense system. Therefore, we measured Nrf2-regulated genes and the oxidant/antioxidant enzymes controlled by the Nrf2 pathway in the IHF and AGF groups. With long-term ryegrass supplementation, the Nrf2-regulated genes *NQO1*, *Gclc*, *Gclm*, and *GSTA4* were higher in the AGF group than those in the IHF group (Appendix A). In addition, we detected higher protein levels of HO-1, GSR, T-SOD, GSH-Px, T-AOC, and CAT in the AGF group than in the IHF group (Figure 6G–L). To validate our results, we determined the mRNA expression levels of *HMOX1* (*HO-1*), *SOD*, *GSH-Px*, and *CAT* from the spleen tissues of the AGF and IHF groups. Again, we identified that ryegrass was more involved in the upregulation of *HMOX1* (*HO-1*), *SOD*, *GSH-Px*, and *CAT* in the AGF group than in the IHF group (Appendix A–E). Furthermore, to identify whether these antioxidants control oxidative mediators that cause ROS/HDAC/Keap1 insults in the AGF and IHF groups, we determined protein levels of MDA from the spleen tissues. We identified higher expression of protein levels of MDA in the IHF group compared to that in the AGF group (Figure 6M). 

### 3.6. Ryegrass-Dependent Keap1-Nrf2 Pathway Activation Impedes Endotoxemia, Systemic Inflammation, and Spleen Dysfunctions

To further explore the effects of ryegrass on gut microbial alterations that facilitate spleen functions at 45 d, 60 d, and 90 d, we hypothesized that gut microbial SCFAs could activate the Nrf2 pathway and mimic oxidative stress-induced systemic inflammation through the gut–microbiota–spleen axis in geese. Carsetti et al. [17] described that circulating IgM can influence the production of SIgA in the gut and thus directly involve the mechanism of a gut–spleen axis. Therefore, we measured the protein levels of IgA, IgM, and IgG with ELISA kits from the caecal tissues, followed by serum samples and spleen tissues. We identified a higher concentration of IgA, IgM, and IgG in caecal tissues, serum samples, and spleen tissues of AGF geese compared to that in IHF geese, suggesting that the circulating immunoglobulins may contribute to spleen functions by influencing the production of IgA, IgM, and IgG in spleen tissues (Figure 7A–I). Furthermore, SCFAs, particularly butyric, acetic, and propionic acids, encourage the Nrf2 redox signaling pathway [38] and regulatory T cells to release *IL-10*, which enters the gut–spleen axis by patrolling CD8 T cells and ameliorating severe cytotoxicity and inflammation [36,39]. To explore the role of anti-inflammatory cytokines in controlling inflammation, we measured the mRNA levels of *Nrf2*, *IL-4*, and *IL-10* and the protein levels of HO-1, GSR, T-SOD, GSH-Px, T-AOC, and CAT from spleen tissues. Compared to gut microbial SCFAs, *Nrf2*, and antioxidant concentrations, the mRNA expression levels of *IL-4* and *IL-10* were higher in the spleen tissues of the AGF group than in the IHF group (Figure 7J,K). 

To further prove the gut–microbiota–spleen axis, we performed a correlation analysis of host markers and gut microbial SCFAs. The results obtained from the correlation of host markers showed that acetic, butyric, propionic, valeric, and isovaleric acids were significantly positively correlated with *Nrf2*, antioxidant enzymes (HO-1, GSR, T-SOD, GSH-Px, T-AOC, and CAT), B cells (*B220*, *CD19*, and *IgD*), T cells (*CD3*, *CD4*, *CD8*, and *IL-2*), immunoglobulins (IgA, IgM, and IgG), and anti-inflammatory cytokines (*IL-4* and *IL-10*) in the AGF group compared with the IHF group (Figure 8A). In contrast, the gut microbial SCFA-induced Nrf2 redox signaling enzymes were more negatively correlated with endotoxemia (LPS), oxidative stress (ROS), *NF-κB*, *NLRP3*, and inflammatory cytokines in the AGF group compared with the IHF group. These results indicated that the existence of dietary fiber in ryegrass is the main influencer of gut microbial metabolic modulations that, on the one hand, promote the redox signaling pathway and immune balance and, on the other hand, attenuate systemic inflammation in geese. 

### 3.7. Long-Term Ryegrass Supplementation Impedes Splenic Dysfunctions by Maintaining the Gut Microbial Microenvironment

The intestinal microbiota plays an important role in the development of the spleen [16,17,18]. How diet affects spleen maturation by modulating gut microbiota needs to be explained in geese. For this consensus, we developed a correlation analysis among the gut microbes (Figure 8B) and with the host markers during the different growth phases of geese (Figure 8C and Appendix A). As shown in Figure 8C and Appendix A, *Eggerthellaceae*, *Oscillospiraceae*, *Rikenellaceae*, and *Lachnospiraceae* were the most abundant bacteria that existed in the caecal chyme of the AGF group that were positively correlated with SCFAs, Nrf2 redox signaling enzymes, B cells, T cells (with an exception of *IL-17* and *TGF-β*), and anti-inflammatory mediators in spleen tissues of geese. In contrast, *Enterococcaceae* and *Prevotellaceae* were highly abundant in the IHF group compared with the AGF group, which were positively correlated with LPS, ROS, *NF-κB*, *NLRP3*, inflammatory cytokines, and spleen weight in the spleen tissues of geese (Figure 8C and Appendix A, and Appendix A). In conclusion, these results describe the role of ryegrass as a high-dietary-fiber source that controls LPS-induced oxidative stress-facilitated systemic inflammation by activating the SCFAs/Nrf2 pathway in geese.

## 4. Discussion

Low-dietary-fiber-dependent gut microbial LPS-induced oxidative stress [2] undergoes systemic inflammation in the spleen [6] and inflammatory diseases [40]. Today, the current poultry industry is based on low-dietary-fiber-enriched feed ingredients [30], which may lead to higher growth performance with compromised intestinal infections, low immunity, and metabolic diseases [31]. In addition, the global trend of consuming a low-dietary-fiber diet has been concerned with improved autoimmunity, diabetes, cardiovascular health, and cancer [22]. Despite this, more research on geese is required to determine how dietary fiber content lowers the risk of inflammatory disorders by triggering a redox signaling pathway that is dependent on SCFAs. Thus, for the first time, the present research explores that ryegrass intake can ameliorate commercial diet-dependent gut microbial alterations, endotoxemia, oxidative stress, and inflammatory diseases. 

However, fewer findings have explained the relationship of altered gut microbiota with spleen functions in humans [41], mice [18], chickens [24], and tilapia [42]. Within these abrogative findings, the mechanisms linked with the diverse association of dietary fiber intervention with the gut–microbiota–spleen axis have not been explored. Notably, we developed a ryegrass-dependent artificial pasture grazing system that concords with concomitant mechanisms to attenuate commercial diet-dependent intestinal and splenic LPS and inflammatory disorders. Subsequently, the advanced role of ryegrass is to develop a pathway-based mechanism, thereby restoring the enrichment of intestinal SCFA-generating microbes and impeding commercial diet-induced LPS-producing bacteria. These modifications improve the Nrf2 redox signaling pathway, splenocyte population (B cells and T cells), immunoglobulins, and anti-inflammatory cytokines, resulting in decreased metabolic endotoxemia and LPS-induced oxidative stress in the spleen organ of geese. The reduction in pro-inflammatory cytokines results in a decrease in chronic inflammatory disorders. In light of the aforementioned data, Spearman correlation analysis provides substantial support for the suggested mechanisms.

Only a few studies explain the character of altered gut microbiota and LPS levels in spleen dysfunctions [43,44]. In this study, we discovered altered microbiota using predicted functions based on 16S rRNA sequencing and Tax4Fun analysis that were involved in LPS-producing strains. Some of the bacteria, *Ruminococcaceae*, *norank_o__RF39*, *Enterocococcaceae*, *norank_o__Clostridia_UCG-014*, *Eubacterium_coprostanoligenes_group*, and *Erysipelatoclostridiaceae,* were strongly involved in KEGG orthologues of LPS production in the IHF group. In contrast, our findings were underscored by those of dietary capsaicin in human participants [45]. This suggests that the reduced enrichment of KEGG orthologues involved in LPS production may be due to the higher abundances of *Oscillospiraceae*, *Eggerthellaceae*, and *Bacteroidaceae* in the AGF group. 

LPS-induced ROS production is mainly due to the stimulation of the TLR4/MyD88 pathway [46]. Several research studies have demonstrated that the stimulation of the TLR4/MyD88 pathway is influenced by a low-dietary-fiber diet and high carbohydrates, lipids, protein, and calories [2,47,48]. Contrary to these studies, we developed the LPS-induced NADPH pathway and then hypothesized that this pathway should promote ROS production in commercial diet-feeding geese (Figure 9). It is known that SrcK activates PI3K in response to LPS, and then PI3K stimulates NADPH oxidase through p47^phox^ phosphorylation in RAW264.7 macrophages [49]. Concerning this study, we detected higher protein concentrations of LPS and mRNA expression levels of *Srck*, *PI3K*, and *p47^phox^* in the IHF group compared with those of the AGF group. P47^phox^ is a major driving factor of NADPH oxidase that is involved in TLR-MyD88 activation [50]. Of note, we detected increased mRNA levels of *NADPH*, *TLR2*, *TLR4*, and *MyD88* in the IHF group compared with the AGF group. According to Ichikawa et al. [46], the activated RLT4/MyD88 pathway further contributes to the production of ROS. Considering this report, we identified higher concentrations of ROS in the serum and spleen tissues of the IHF group compared with those of the AGF group. 

The production of oxidative stress due to ROS insults provides an environment for the activation of the NF-κB pathway [51], which plays a vital role in the establishment of systemic inflammation through the maturation of pro-inflammatory cytokines [52,53]. Our previous work had limitations, whereby we showed that the development of inflammation by the LPS/ROS-induced NF-κB pathway was *LC8-* and *IkB-a-*dependent [2]. However, in the current study, we showed a mechanistic pathway by which the first NF-κB pathway activates *NLRP3* and then allows it to induce pro-inflammatory cytokine stimulation. 

In many studies, the augmentation of the *NLRP3* inflammasome is *NF-κB*-dependent [52]. However, in our investigation, we identified that the stimulation of the *NLRP3* inflammasome by *NF-κB* is MyD88-dependent *TRAF6* regulation. Several studies using LPS as an *MyD88* activator support our findings [54]. Furthermore, our investigations, somewhat modified from those of Lu et al. [55] and Sato et al. [56], demonstrated that *MyD88* triggering promoted *IRAK*, *TRAF6*, and *TAK1* activation, which might have collectively activated *NF-κB* signaling cascades and finally resulted in *NLRP3* activation in the IHF group. Several studies demonstrated that the overexpression of *NLRP3* results in increased *caspase-1*, which matures pro-inflammatory cytokines *IL-1β* and *IL-18* [57,58] and then influences other inflammatory cytokines [59,60]. Comparing these studies with our current work, we identified higher mRNA expression levels of *NLRP3*, *Casoase-1*, and pro-inflammatory cytokines in the IHF group compared to those of the AGF group. A defensive role of the spleen against inflammation and obesity-related diseases has been reported elsewhere [61,62]. How dietary interventions impact immune responses of the spleen against inflammatory mediators is not reported clearly in geese. Herein, comparing the expression levels of inflammatory markers, we detected a parallel decrease in the expression levels of splenocyte populations (B cells and T cells) except *IL-17*, *TGF-β*, and spleen weight in the IHF group compared with those of the AGF group.

The perpetuated regulation of HDACs and NF-κB is primarily concerned with the activation of genes that are commonly expressed during inflammation [63,64]. In our previous study, we described the mechanism by which the AGF system-induced alkaline phosphatase enzyme can dephosphorylate LPS and attenuate the existence of NF-κB-facilitated chronic low-grade inflammation in the intestines of geese. In this study, we developed a pathway by which ryegrass-dependent gut microbial SCFA receptors activate the AMPK pathway and then collectively inhibit HDAC. Upon HDAC inhibition, Keap1 dissociates from Nrf2 and then enters the activation of ARE-dependent genes in the spleen tissues of the AGF group. Our findings were followed by the reports of Wang et al. [37], in which we detected higher concentrations of SCFA-producing bacteria and mRNA expressions of SCFAs receptors *GPCR109A*, *FFAR2*, *FFAR3*, and *AMPK* that attenuated the severity of *HDAC* in the AGF group compared to those of the IHF group. Additionally, according to the reports of Xu et al. [65] and Xu et al. [66], we detected a significant difference between the expression levels of *Keap1*, *Nrf2*, and Nrf2-regulated genes, describing the important role of ryegrass in activating the Nrf2 redox signaling pathway in geese. 

It is well known that Nrf2 increases the ARE-dependent gene expression of a series of antioxidant proteins [67] and regulatory T cells to release anti-inflammatory cytokines [68], which enter the gut–spleen axis by patrolling CD8 T cells and ameliorating severe cytotoxicity and inflammation [36,39]. To further investigate the function of gut microbial SCFAs in controlling spleen functions, we performed a correlation analysis among host markers and gut microbial SCFAs (caecum vs. spleen). We described, for the first time, that the SCFAs were significantly positively correlated with Nrf2 redox signaling enzymes, T cells (except *IL-17* and *TFG-β*), B cells, immunoglobulins, and anti-inflammatory mediators. We can say that this is our first report in which long-term ryegrass-induced gut microbial SCFAs, along with immunoglobulins, *IL-4,* and *IL-10* expression, can collectively attenuate the LPS-induced oxidative stress and systemic inflammation in geese.

The gut microbiota controls the maturation of the spleen [16,17,18]. To determine how diet affects spleen maturation by modulating the gut microbiota, we performed a correlation analysis among the gut microbes and with the host markers during the different growth phases of geese. Herein, the most notable findings were the higher abundance of *Eggerthellaceae*, *Oscillospiraceae*, and *Lachnospiraceae* in the AGF group that were positively correlated with SCFAs, Nrf2 redox signaling enzymes, B cells, T cells (with the exception of *IL-17* and *TGF-β*), and anti-inflammatory mediators in the spleen tissues of geese. A well-known SCFA-producing *Oscillospiraceae* is known to treat obesity-related diseases [69,70]. Similarly, the role of *Lachnospiraceae* as probiotics has been acknowledged in attenuating colonic dysbiosis-related metabolic syndrome and cancer [71]. In contrast, ryegrass ameliorated the higher abundances of *Enterococcaceae* and *Prevotellaceae* that were positively correlated with LPS, ROS, *NF-κB*, *NLRP3*, inflammatory cytokines, and spleen weight in the spleen tissues of IHF geese. Few studies have shown that the higher abundance of *Prevotellaceae* causes myocardial ischemia [8] and intestinal inflammation [72] in rats. In contrast, *Enterococcaceae* is known as a multidrug-resistant nosocomial pathogen that causes roughly 66,000 infections/year in the USA [73]. It primarily causes septic arthritis and meningitis in immunocompromised patients [74]. In conclusion, these results describe the role of ryegrass as a high-dietary-fiber source that attenuates systemic inflammation-producing bacteria by activating SCFA-producing bacteria in geese.

## 5. Conclusions

In conclusion, gut microbial SCFA production by ryegrass supplementation appears to restore gut microbial architecture, which then promotes Nrf2 redox signaling enzymes, B cells, and regulatory T cells to release anti-inflammatory cytokines and immunoglobulins in the spleens of geese. In addition, the higher abundances of these markers seemed to attenuate endotoxemia, oxidative stress, and systemic inflammation in the spleens of commercial diet-fed geese through the gut–microbiota–spleen axis.

## Figures and Tables

**Figure 1 nutrients-16-00747-f001:**
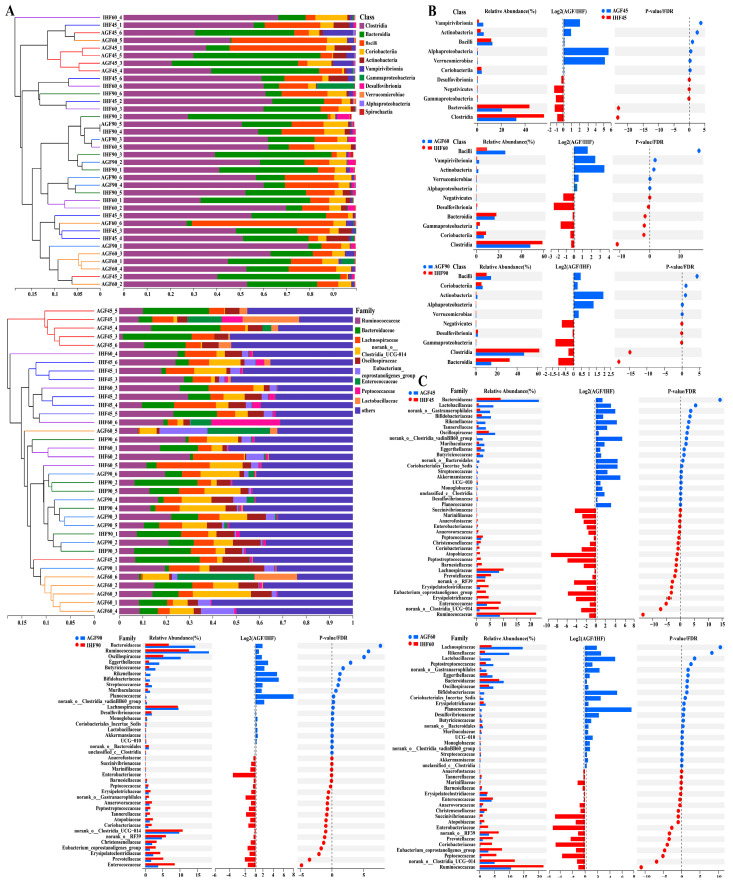
Effects of ryegrass on the gut microbial composition of geese. (**A**) Bar chart for the bacterial community composition at class and family levels. (**B**) Differences in bacterial abundance between the two groups at the class level (statistical results are described in Appendix A). (**C**) Differences in bacterial abundance between the two groups at the family level (statistical results are explained in Appendix A).

**Figure 2 nutrients-16-00747-f002:**
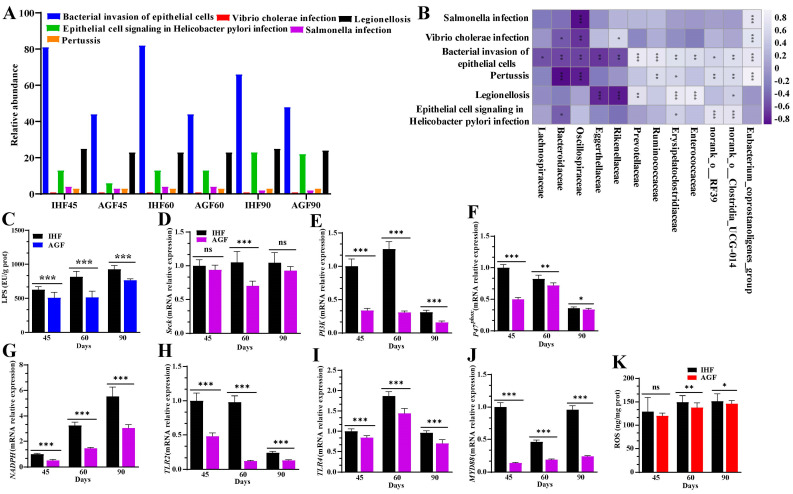
Commercial diet-dependent gut microbial alterations are paralleled by LPS-induced oxidative stress production. (**A**) Relative abundance of KEEG orthologues involved in LPS biosynthesizing-related functions (statistical results are described in Appendix A). (**B**) Relative contributions of various species to the gut microbiome’s overall LPS-encoding ability. (**C**) LPS concentration in spleen tissues. (**D**) *Srck* mRNA relative expression in spleen tissues. (**E**) *PI3k* mRNA relative expression in spleen tissues. (**F**) *P47^phox^* mRNA relative expression in spleen tissues. (**G**) *NADPH* mRNA relative expression in spleen tissues. (**H**) *TLR2* mRNA relative expression in spleen tissues. (**I**) *TLR4* mRNA relative expression in spleen tissues. (**J**) *MyD88* mRNA relative expression in spleen tissues. (**K**) ROS concentration in spleen tissues. In-house feeding system (IHF) and artificial pasture grazing system (AGF). Data with ** p* < 0.05, *** p* < 0.01, and **** p* < 0.001 are considered significant; ns: not significant (Student’s *t*-test, *p* < 0.05).

**Figure 3 nutrients-16-00747-f003:**
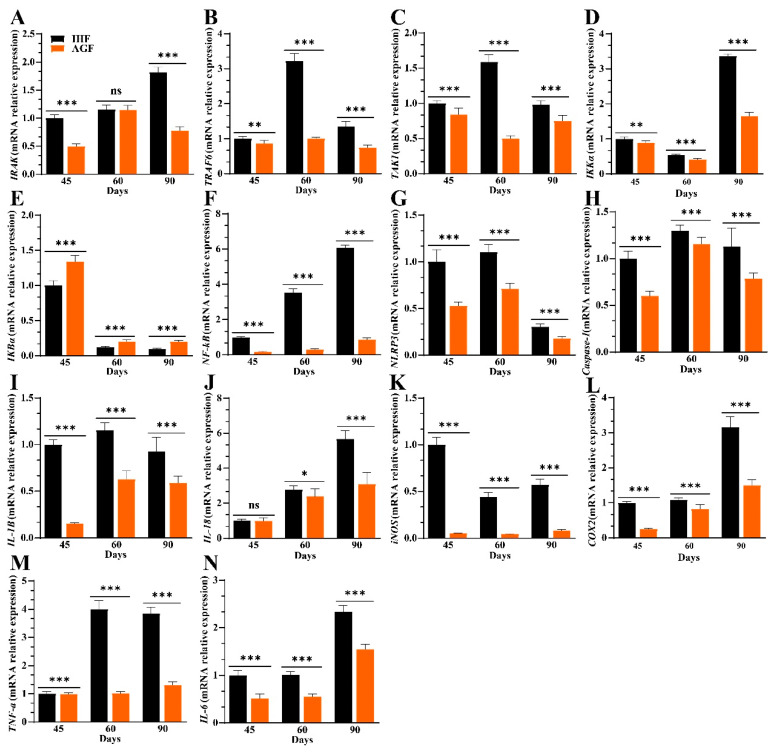
Commercial diet-induced oxidative stress causes NF-κB/NLRP3 pathway-induced systemic inflammation. (**A**) *IRAK* mRNA relative expression in spleen tissues. (**B**) *TRAF6* mRNA relative expression in spleen tissues. (**C**) *TAK1* mRNA relative expression in spleen tissues. (**D**) *IKKα* mRNA relative expression in spleen tissues. (**E**) *IKBα* mRNA relative expression in spleen tissues. (**F**) *NF-κB* mRNA relative expression in spleen tissues. (**G**) *NLRP3* mRNA relative expression in spleen tissues. (**H**) *Caspase-1* mRNA relative expression in spleen tissues. (**I**) *IL-1β* mRNA relative expression in spleen tissues. (**J**) *IL-18* mRNA relative expression in spleen tissues. (**K**) *iNOS* mRNA relative expression in spleen tissues. (**L**) *COX2* mRNA relative expression in spleen tissues. (**M**) *TNF-α* mRNA relative expression in spleen tissues. (**N**) *IL-6* mRNA relative expression in spleen tissues. In-house feeding system (IHF) and artificial pasture grazing system (AGF). Data with ** p* < 0.05, *** p* < 0.01, and **** p* < 0.001 were considered significant; ns: not significant (Student’s *t*-test, *p* < 0.05).

**Figure 4 nutrients-16-00747-f004:**
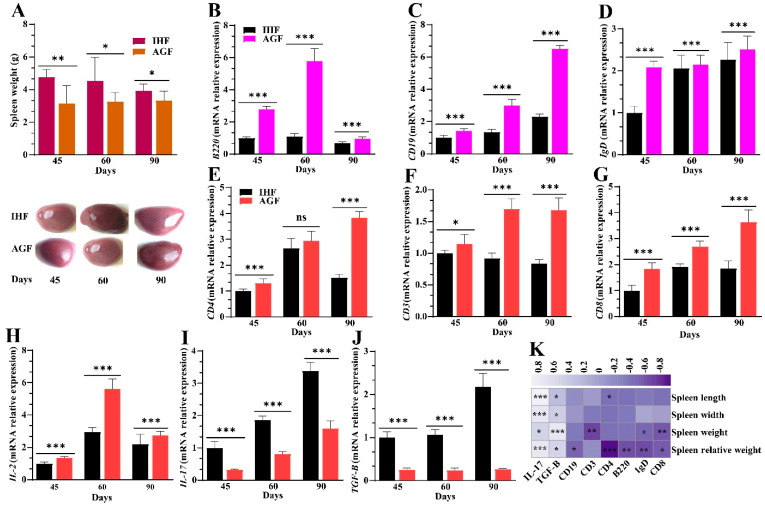
Commercial diet-induced systemic inflammation increases spleen weight and modulates splenocyte populations. (**A**) Spleen weight. (**B**) *B220* mRNA relative expression in spleen tissues. (**C**) *CD19* mRNA relative expression in spleen tissues. (**D**) *IgD* mRNA relative expression in spleen tissues. (**E**) *CD4* mRNA relative expression in spleen tissues. (**F**) *CD3* mRNA relative expression in spleen tissues. (**G**) *CD8* mRNA relative expression in spleen tissues. (**H**) *IL-2* mRNA relative expression in spleen tissues. (**I**) *IL-17* mRNA relative expression in spleen tissues. (**J**) *TGF-β* mRNA relative expression in spleen tissues. (**K**) Spearman correlation analysis between spleen weight, spleen relative weight, spleen length, spleen width, and splenocyte populations. Purple squares indicate a negative correlation, and white squares indicate a positive correlation. A deeper color describes a stronger correlation. In-house feeding system (IHF) and artificial pasture grazing system (AGF). Data with ** p* < 0.05, *** p* < 0.01, and **** p* < 0.001 are considered significant; ns: not significant (Student’s *t*-test, *p* < 0.05).

**Figure 5 nutrients-16-00747-f005:**
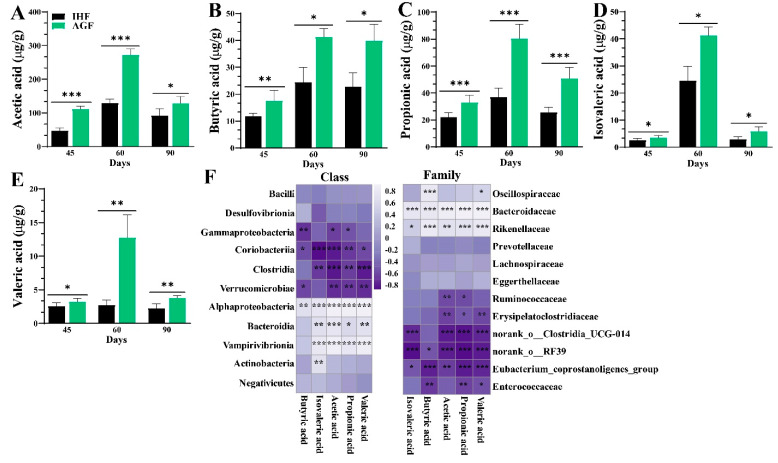
Effect of long-term ryegrass intake on gut microbial short-chain fatty acids. (**A**–**E**) Measurement of acetic acid, butyric acid, propionic acid, valeric acid, and isovaleric acid from caecal chyme. (**F**) Spearman correlation analysis of SCFAs with bacterial abundance at the class and family levels (statistical results are described in Appendix A). Purple squares indicate a negative correlation, and white squares indicate a positive correlation. A deeper color describes a stronger correlation. In-house feeding system (IHF) and artificial pasture grazing system (AGF). Data with ** p* < 0.05, *** p* < 0.01, and **** p* < 0.001 are considered significant (Student’s *t*-test, *p* < 0.05).

**Figure 6 nutrients-16-00747-f006:**
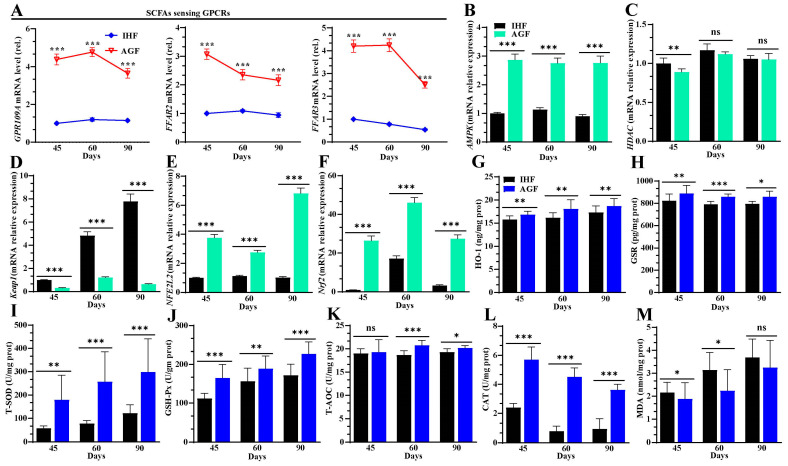
Ryegrass-dependent SCFA regulation improved Keap1-Nrf2 pathway activation. (**A**) mRNA relative expression of GPCRs (*GPR109A*, *FFAR2*, and *FFAR3*) in spleen tissues. (**B**) *AMPK* mRNA relative expression in spleen tissues. (**C**) *HDAC* mRNA relative expression in spleen tissues. (**D**) *Keap1* mRNA relative expression in spleen tissues. (E) *NFE2L2* mRNA relative expression in spleen tissues. (**F**) *Nrf2* mRNA relative expression in spleen tissues. (**G**–**M**) HO-1, GSR, T-SOD, GSH-Px, T-AOC, CAT, and MDA protein levels were measured from spleen tissues using ELISA kits. In-house feeding system (IHF) and artificial pasture grazing system (AGF). Data with ** p* < 0.05, *** p* < 0.01, and **** p* < 0.001 are considered significant; ns: not significant (Student’s *t*-test, *p* < 0.05).

**Figure 7 nutrients-16-00747-f007:**
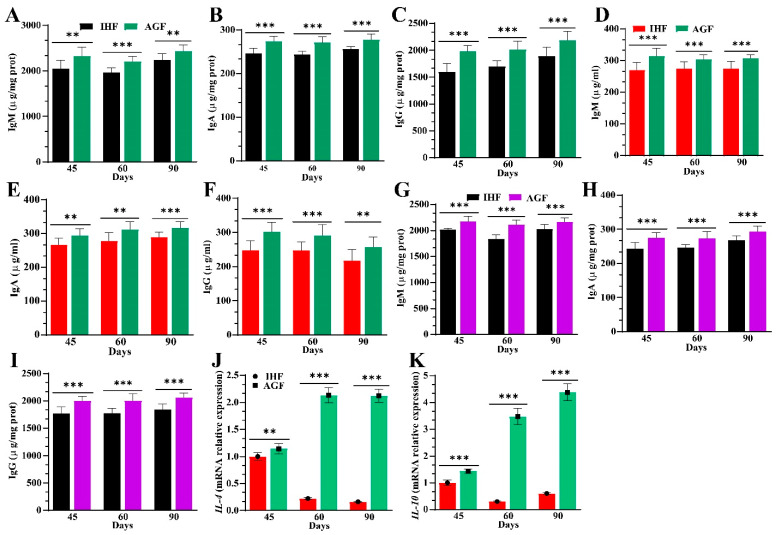
Effects of ryegrass on immunoglobulins and anti-inflammatory cytokine production. (**A**–**C**) IgM, IgA, and IgG protein levels measurement from caecal tissues. (**D**–**F**) IgM, IgA, and IgG protein levels measurement from serum samples. (**G**–**I**) IgM, IgA, and IgG protein levels measurement from spleen tissues. (**J**) *IL-4* mRNA relative expression in spleen tissues. (**K**) *IL-10* mRNA relative expression in spleen tissues. In-house feeding system (**I**,**H**,**F**) and artificial pasture grazing system (**A**,**G**,**F**). Data with ** *p* < 0.01, and *** *p* < 0.001 are considered significant (Student’s *t*-test, *p* < 0.05).

**Figure 8 nutrients-16-00747-f008:**
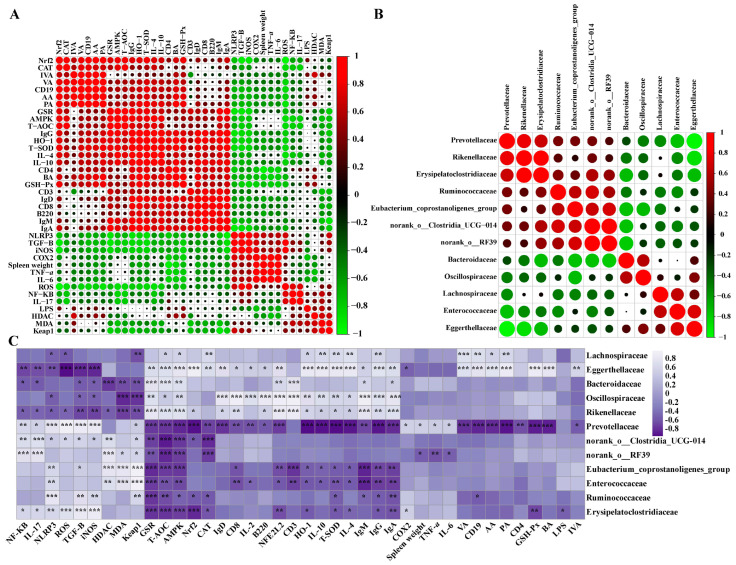
Effect of ryegrass supplementation on ameliorating splenic dysfunctions by maintaining gut microbial microenvironment. (**A**) A Spearman correlation analysis among host markers. (**B**) A Spearman correlation analysis among highly abundant gut microbiota at the family level. (**C**) A Spearman correlation analysis of host markers and gut microbiota at the family level (statistical results are described in Appendix A). Data with ** p* < 0.05, *** p* < 0.01, and **** p* < 0.001 are considered significant (Student’s *t*-test, *p* < 0.05).

**Figure 9 nutrients-16-00747-f009:**
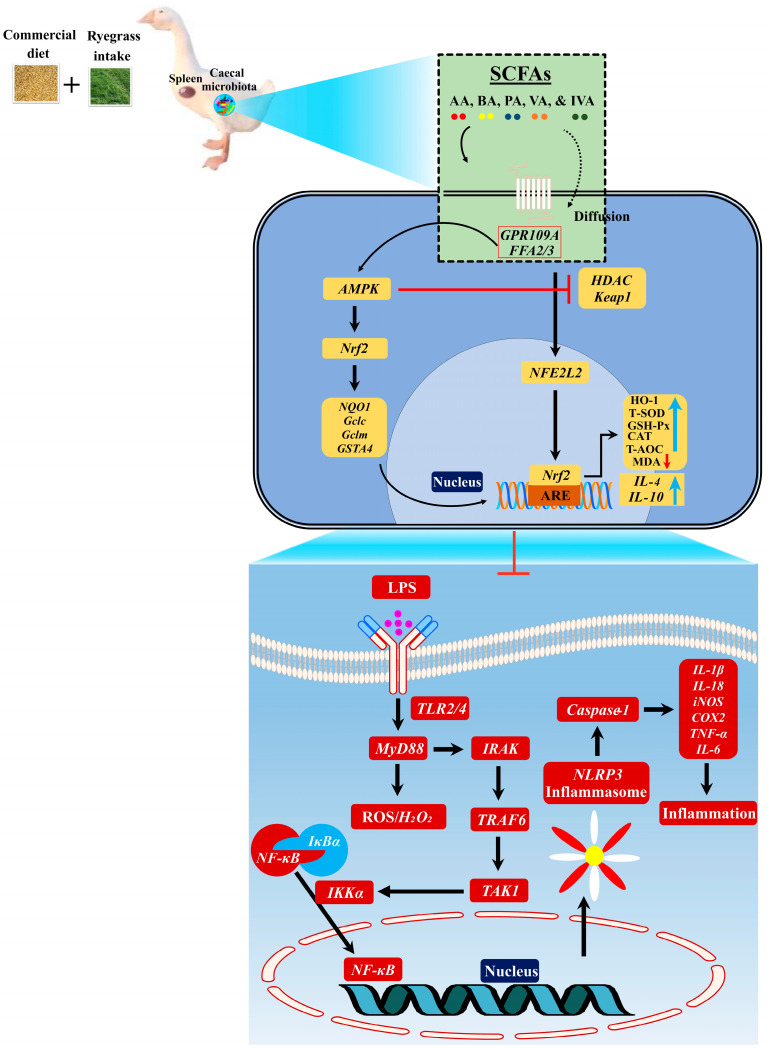
Graphical illustration of the interaction between the gut microbial SCFA-induced Nrf2 redox signaling pathway and LPS/ROS facilitated systemic inflammatory cytokines. Ryegrass upregulates SCFAs and SCFA-producing bacteria, which enhances the GPCR-induced AMPK pathway. This pathway causes the dissociation of Keap1 and Nrf2, dependent on HDAC inhibition, and then regulates the Nrf2 signaling pathway. Next, the increased concentrations of redox signaling Nrf2 pathway-regulated enzymes such as HO-1, T-SOD, GSH-Px, CAT, and T-AOC with an exception of MDA and anti-inflammatory cytokines (*IL-4* and *IL-10*) collectively ameliorate LPS-induced ROS-mediated NF-κB/NLRP3 inflammasome formation. The downstream regulation of NF-κB/NLRP3 inflammasome formation resulted in reduced systemic inflammation in the spleen tissues of AGF geese. AA, acetic acid; BA, butyric acid; PA, propionic acid; VA, valeric acid; IVA, isovaleric acid.

## Data Availability

Data are contained within the article and Appendix A.

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
