# Peer review of "Supplementing Ryegrass Ameliorates Commercial Diet-Induced Gut Microbial Dysbiosis-Associated Spleen Dysfunctions by Gut–Microbiota–Spleen Axis"

_nutrients, 2024, doi:10.3390/nu16050747_

Round 1
Reviewer 1 Report
Comments and Suggestions for Authors
Furthermore, the abstract is not written in the format of an abstract in terms of detail in methodology, e.g. there are no mention of geese, nor does the introduction prepare the reader for the content of the manuscript later on. The results are presented in a non-objective manor and are written at times like a discussion. It is difficult to discern which statement relate to the results of the current study or previous literature. There is a lot of work and significant results presented but the manuscript does not do it justice. There is a lot of talk in the discussion about pathways, however many events in these pathways are post-translational and not captured using transcriptional methods. Images of pathways would be useful in terms of representing the findings in this context. The expression of genes being measured are not definitive markers of the cell types being discussed yet the discussion is written in a way that assumes they are. The results and methodology may be sound but this is not being captured in the writing.
Comments on the Quality of English LanguageThis manuscript needs significant editing. There are grammatical errors through-out in addition to statements that are vague and terminology that is unscientific.
Author Response
Reviewer 1
Furthermore, the abstract is not written in the format of an abstract in terms of detail in methodology, e.g. there are no mention of geese, nor does the introduction prepare the reader for the content of the manuscript later on. The results are presented in a non-objective manor and are written at times like a discussion. It is difficult to discern which statement relate to the results of the current study or previous literature. There is a lot of work and significant results presented but the manuscript does not do it justice. There is a lot of talk in the discussion about pathways, however many events in these pathways are post-translational and not captured using transcriptional methods. Images of pathways would be useful in terms of representing the findings in this context. The expression of genes being measured are not definitive markers of the cell types being discussed yet the discussion is written in a way that assumes they are. The results and methodology may be sound but this is not being captured in the writing.
This manuscript needs significant editing. There are grammatical errors through-out in addition to statements that are vague and terminology that is unscientific.
Q1: Furthermore, the abstract is not written in the format of an abstract in terms of detail in methodology, e.g. there are no mention of geese, nor does the introduction prepare the reader for the content of the manuscript later on.
Response 1: Respected reviewer, we have briefly explained the ABSTRACT according to journal formatting in terms of methodology with mostly emphasis on geese (highlighted with yellow color), and the newly updated ABSTRACT has been highlighted with green color. Please find similar changes in the abstract of the article and herein.
Therefore, the gut microbial composition of the caecal chyme of geese was assessed by 16S rRNA microbial genes, and Tax4Fun analysis was revealed to identify the enrichment of KEGG orthologues involved in lipopolysaccharide production. The concentrations of LPS, reactive oxygen species, antioxidant/oxidant enzymes, and immunoglobulins were measured from serum samples and spleen tissues by ELISA kits. Quantitative reverse transcription PCR was employed to detect the Kelch-like-ECH-associated protein 1-Nuclear factor erythroid 2-related factor 2 (Keap1-Nrf2), B cells and T cells targeting markers, and anti-inflammatory/inflammatory cytokines from the spleen tissues of geese. The SCFAs were determined from the caecal chyme of geese by using gas chromatography.
In addition, we have updated the aim of INTRODUCTION to make it clear for readers and highlighted it with yellow color according to your suggestions. For methodology, please find the updated materials and methods section.
Based on previous reports that the gut-microbiota-spleen axis modulates systemic inflammation-related processes [18,19,33], we aimed to investigate the effects of ryegrass on attenuating the gut microbial alterations-associated spleen dysfunctions in geese. In this context, first, we developed a mechanistic pathway in which ryegrass, upon restoring SCFAs-producing gut microbiota, activates the Nrf2 redox signaling pathway in the spleen organs of geese. Then, it was investigated that this pathway elicits B cells and regulatory T cells to stimulate anti-inflammatory cytokines and immunoglobulins, which alternatively attenuate oxidative stress-induced systemic inflammation in geese. Lastly, a correlation analysis of the gut microbiome with host markers was established to better understand the spleen functions through the gut microbiota-spleen axis.
Q2: The results are presented in a non-objective manor and are written at times like a discussion. It is difficult to discern which statement relate to the results of the current study or previous literature. There is a lot of work and significant results presented but the manuscript does not do it justice.
Response 2: First, we explained our main objectives in the sections of ABSTRACT and INTRODUCTION (highlighted in yellow color). Then objectively, we raised the questions in the RESULT section and tried to solve them (highlighted in yellow color). For more understanding and justification of which statement of result relates to which previous literature, we have created several questions with references, and then according to them, we have tried our best to solve them in the RESULTS section (highlighted in yellow color).
According to the objective manner of old literature and our hypothesis, we have tried our best to justify our manuscript. For more details, please see the highlighted section of DISCUSSION (highlighted with yellow color). All the updated work is highlighted in green and yellow colors.
Q 3: There is a lot of talk in the discussion about pathways, however many events in these pathways are post-translational and not captured using transcriptional methods. Images of pathways would be useful in terms of representing the findings in this context. The expression of genes being measured are not definitive markers of the cell types being discussed yet the discussion is written in a way that assumes they are.
Response 3: Respectfully, we agree with you that most of the events in the pathways explained in the discussion section are post-translational. Indeed, according to animal and human studies, we are representing this report for the first time, that’s why to confirm them, we have selected the post-translational events in this study. The full names of terms regarding the genes have been highlighted with green color in the RESULT section. Furthermore, to justify the images of pathways, we have provided most of the literature with references in the RESULT and DISCUSSION sections (green and yellow colors).
Our findings were followed by the reports of Wang et al. [38], in which, we detected higher concentrations of SCFA-producing bacteria and mRNA expressions of SCFAs receptors GPCR109A, FFAR2, FFAR3, and AMPK that attenuated the severity of HDAC in the AGF group compared to those of the IHF group. Additionally, according to the reports of Xu et al. [67] and Xu et al. [68], we detected a significant difference between the expression levels of Keap1, Nrf2, and Nrf2-regulated genes, describing the important role of ryegrass in activating the Nrf2 redox signaling pathway in geese. [Note: for more evidence, we have explained the diet-dependent activation of SCFAs, their receptors, the Keap1-Nrf2 signaling pathway (gene expression and protein levels of oxidants and antioxidants enzymes), and the mechanisms of SCFAs-activated Nrf2 pathway in attenuating systemic inflammation (RESULT AND DISCUSSION, sections)].
For more information, please see the legend of Figure 9.
Figure 9. Graphical illustration of the interaction between the gut microbial SCFA-induced Nrf2 redox signaling pathway and LPS/ROS facilitated systemic inflammatory cytokines. Ryegrass upregulates SCFAs and SCFA-producing bacteria, which enhances the GPCRs-induced AMPK pathway. This pathway causes the dissociation of Keap1 and Nrf2, dependent on HDAC inhibition, and then regulates the Nrf2 signaling pathway. Next, the redox signaling Nrf2 pathway and anti-inflammatory cytokines collectively ameliorate LPS-induced ROS-mediated NF-κB/NLRP3 inflammasome formation. The downstream regulation of NF-κB/NLRP3 inflammasome formation resulted in reduced systemic inflammation in the spleen tissues of AGF geese. AA, acetic acid; BA, butyric acid; PA, propionic acid; VA, valeric acid; and IVA, isovaleric acid.
We agree with you that we have measured the expression of genes for cell types. But to justify the functions of cell types, we have determined the protein levels of immunoglobulins (IgA, IgG, and IgM), antioxidants (HO-1, GSR, T-SOD, GSH-Px, T-AOC, and CAT), and oxidants (MDA). Furthermore, the gut microbial composition of caecal chyme of geese was assessed by 16S rRNA microbial genes and the SCFAs were determined through gas chromatography. According to previous studies, intestinal microbiota plays an important role in the development of the spleen [references; 16-18], see RESULT sections (3.6. Ryegrass-dependent Keap1-Nrf2 pathway activation impedes endotoxemia, systemic inflammation, and spleen dysfunctions, and 3.7. Long-term ryegrass supplementation impedes splenic dysfunctions by maintaining the gut microbial microenvironment, (highlighted with yellow color)), we did a correlation of gut microbiome and host markers (mRNA expressed genes and their protein levels). Then we explained this part in the DISCUSSION section briefly in the form of Figure 9 along with context (highlighted with yellow color).
Q 4: The results and methodology may be sound but this is not being captured in the writing.
Response 4: Respected reviewer, we appreciate your suggestions. We have improved and updated the sections of the results and methodology and highlighted them with yellow and green colors.
Q 5: This manuscript needs significant editing. There are grammatical errors through-out in addition to statements that are vague and terminology that is unscientific.
Response 5: Respected reviewer, we appreciate your kind suggestion. We have significantly edited the whole manuscript from a professional person. We have added new scientific terminologies and have removed grammatical errors throughout the manuscript (highlighted with yellow and green colors).
Reviewer 2 Report
Comments and Suggestions for Authors
This manuscript titled “Supplementing Ryegrass Ameliorates Commercial Diet-Induced Gut Microbial Dysbiosis-Associated Spleen Dysfunctions by Gut-Microbiota-Spleen Axis” described positive effects of Ryegrass on gut microbiome of geese.
- From only one experimental data, it is very difficult to give any fidelity to whether this result can be replicated. However, based on this experiment’s data, the manuscript is well organized and very informative.
- In “ The protein concentrations of LPS” Laps is not protein.
- Figure 8 was missed in text.
-In the legend of Figure 9, AGF meat geese may be AGF geese.
Author Response
Reviewer 2
This manuscript titled “Supplementing Ryegrass Ameliorates Commercial Diet-Induced Gut Microbial Dysbiosis-Associated Spleen Dysfunctions by Gut-Microbiota-Spleen Axis” described positive effects of Ryegrass on gut microbiome of geese.
- From only one experimental data, it is very difficult to give any fidelity to whether this result can be replicated. However, based on this experiment’s data, the manuscript is well organized and very informative.
Response 1: The experiment has lasted for 90 days including 24 days of brooding period. The samples from six geese per group were collected on 45d, 60d, and 90d (see Figure S1). We think that the results based on experimental data would be appreciable for further processing.
- In “ The protein concentrations of LPS” Laps is not protein.
Response 2: Respected reviewer, we appreciate your affectionate suggestion. We have corrected this mistake from all over the manuscript and highlighted it with green color.
- Figure 8 was missed in text.
Response 3: Respected reviewer, thank you for correcting us. We have added Figure 8 to the manuscript.
-In the legend of Figure 9, AGF meat geese may be AGF geese.
Response 4: Respected reviewer, thank you for correcting our mistake. We have replaced AGF meat geese with AGF geese in the legend of Figure 9 and highlighted it with a yellow color.
Reviewer 3 Report
Comments and Suggestions for Authors
Dear Authors,
I believe that the manuscript should carefully establish and present the research hypothesis and the purpose of your research. Moreover, the introduction is written in a vague manner and does not prove why such research was undertaken or what the scientific premises were. Therefore, it is not known on what basis the conclusions were drawn, since the hypothesis and purpose of the research were not provided.
Author Response
Reviewer 3
I believe that the manuscript should carefully establish and present the research hypothesis and the purpose of your research. Moreover, the introduction is written in a vague manner and does not prove why such research was undertaken or what the scientific premises were. Therefore, it is not known on what basis the conclusions were drawn, since the hypothesis and purpose of the research were not provided.
Response 1: Respected reviewer, we are thankful for your kind suggestions. We have improved our introduction part in terms of grammatical errors, hypothesis, and purpose. In brief, we have removed the grammatical errors and have highlighted them in yellow and green colors. In addition, please find the aim of our work below.
Based on previous reports that the gut-microbiota-spleen axis modulates systemic inflammation-related processes [18,19,33], we aimed to investigate the effects of ryegrass on attenuating the gut microbial alterations-associated spleen dysfunctions in geese. In this context, first, we developed a mechanistic pathway in which ryegrass, upon restoring SCFAs-producing gut microbiota, activates the Nrf2 redox signaling pathway in the spleen organs of geese. Then, it was investigated that this pathway elicits B cells and regulatory T cells to stimulate anti-inflammatory cytokines and immunoglobulins, which alternatively attenuate oxidative stress-induced systemic inflammation in geese. Lastly, a correlation analysis of the gut microbiome with host markers was established to better understand the spleen functions through the gut microbiota-spleen axis.
Round 2
Reviewer 1 Report
Comments and Suggestions for Authors
The manuscript has improved and is much clearer
Reviewer 3 Report
Comments and Suggestions for Authors
Dear Authors,
the manuscript was corrected, which significantly improved its quality.